# Fabrication of Gum Arabic-Graphene (GGA) Modified Polyphenylsulfone (PPSU) Mixed Matrix Membranes: A Systematic Evaluation Study for Ultrafiltration (UF) Applications

**DOI:** 10.3390/membranes11070542

**Published:** 2021-07-16

**Authors:** Alaa Mashjel Ali, Khalid T. Rashid, Ali Amer Yahya, Hasan Sh. Majdi, Issam K. Salih, Kamal Yusoh, Qusay F. Alsalhy, Adnan A. AbdulRazak, Alberto Figoli

**Affiliations:** 1Membrane Technology Research Unit, Chemical Engineering Department, University of Technology, Alsinaa Street 52, Baghdad 10066, Iraq; 80066@uotechnology.edu.iq (A.M.A.); 80007@uotechnology.edu.iq (K.T.R.); ali.a.yahya@uotechnology.edu.iq (A.A.Y.); 80036@uotechnology.edu.iq (A.A.A.); 2Department of Chemical Engineering and Petroleum Industries, Al-Mustaqbal University College, Babylon 51001, Iraq; hasanshker1@gmail.com (H.S.M.); Dr_IssamKamil@mustaqbal-college.edu.iq (I.K.S.); 3Department of Chemical Engineering, College of Engineering, University Malaysia Pahang, Pahang 26300, Malaysia; kamal@ump.edu.my; 4Institute on Membrane Technology, National Research Council (ITM-CNR), 87030 Rende (CS), Italy; a.figoli@itm.cnr.it

**Keywords:** Gum Arabic-Graphene, sodium alginate, PPSU, membrane modification, ultrafiltration

## Abstract

In the current work, a Gum, Arabic-modified Graphene (GGA), has been synthesized via a facile green method and employed for the first time as an additive for enhancement of the PPSU ultrafiltration membrane properties. A series of PPSU membranes containing very low (0–0.25) wt.% GGA were prepared, and their chemical structure and morphology were comprehensively investigated through atomic force microscopy (AFM), Fourier transforms infrared spectroscopy (FTIR), *X*-ray diffraction (XRD), and field emission scanning electron microscopy (FESEM). Besides, thermogravimetric analysis (TGA) was harnessed to measure thermal characteristics, while surface hydrophilicity was determined by the contact angle. The PPSU-GGA membrane performance was assessed through volumetric flux, solute flux, and retention of sodium alginate solution as an organic polysaccharide model. Results demonstrated that GGA structure had been successfully synthesized as confirmed XRD patterns. Besides, all membranes prepared using low GGA content could impart enhanced hydrophilic nature and permeation characteristics compared to pristine PPSU membranes. Moreover, greater thermal stability, surface roughness, and a noticeable decline in the mean pore size of the membrane were obtained.

## 1. Introduction

Clean water is a pivotal commodity for sustaining human life on earth along with food and energy. Despite water covering around three-quarters of our planet area, direct access to clean water resources is not as easy a matter as someone might think [1]. Technologies have been harnessed over the past decades to achieve this goal. Although these technologies were able to treat complex water resources, there are still doubts whether the paid environmental and economic costs outweigh their benefits. One of these realistic techniques, which found unlimited versatile applications not only in water treatment but also in a wide spectrum of industries, is membrane technology. In the mid-nineteen-sixties, membrane technology witnessed its first inception by the Loeb–Sourirajan research group. Since that time, a tremendous amount of systematic scientific research has been devoted to getting deeper insights into enhancing membrane performance. This was carried through by playing with fabrication parameters aiming to obtain a membrane with desired selectivity and permeation characteristics. These parameters are directly interlinked with membranes characteristics such as hydrophilicity, pore size and their distribution, roughness, and surface charges [2,3,4,5,6,7]. In this context, an unlimited part of this evolution in the membranes industry was dedicated to modifying these organic membranes through chemical and physical methods. However, with emerging nanomaterials and their commercial mass production during the past few decades, various carbon/metal-based nanomaterials have readily split their way into the membrane fabrication and modification field.

Nanocomposite membranes have stood out as a novel class of membranes, merging the characteristics of both organic polymers and organic/inorganic nanoscale materials to end up with an enhanced membrane performance compared to the standard organic one [8]. A vast literature has reported extraordinary enhancements in hydrophilicity, chemical and thermal stability, permeability, porosity, and mechanical characteristics of these nanocomposite membranes. Indeed, they are believed to bestow tremendous opportunities for the future of water treatment applications [9,10,11]. However, it should be noted that in certain circumstances, impregnation of these nanomaterials on the surface or within the polymeric matrices may arise serious issues in the process and environment. Nanomaterials leaching represents one of these disruptive troubles, especially on the morphology of the nanocomposites and their selectivity indeed. Therefore, adopting the application-specific nanomaterials at optimum composition has a crucial influence on the final nanocomposite membrane performance [10,12,13,14,15,16].

Among the wide range of available nanomaterials, carbon-based derivative nanomaterials such as single and multiwalled carbon nanotubes, fullerenes, and graphene have demonstrated exceptional potentials in water treatment-based applications [17]. It is widely accepted that graphene and its derivatives could bestow favorable features for a variety of industrial applications as a standalone membrane and in combination with organic polymers [18,19,20,21]. Graphene (G) is known as a wonder material of the 21st century. G is composed of two-dimensional, monoatomic-thick building blocks of a carbon allotrope, which impart better intrinsic electrical, thermal, and mechanical properties, greater surface area, and aspect ratio than other carbon-based nanomaterials [22]. Herein, it is worth mentioning that pristine graphene may suffer a lack of compatibility with the polymeric matrix, which stems from its weak interfacial bonding with organic polymers and ultimately induces a nonhomogeneous distribution and inferior membrane properties. Oxidation of graphene has been suggested to revise its surface characteristics and promote its colloidal stability [23,24,25]. The presence of functional groups, such as carboxyl at the edge and epoxide and hydroxyl at the basal plane, bestow the impressive characteristics of graphene oxide (GO). This chemically modified form of GO could indeed impart high colloidal dispersion in organic solvents and compatibility with organic polymers [26]. GO has evinced high electrostatic repulsion against many organic solutes, e.g., Bovine serum albumin (BSA), and performed as a barrier preventing their adsorption [27]. The versatility of GO functional groups presented on the membrane surface is believed to endow magnificent hydrophilicity and antifouling characteristics on that nanocomposite membrane [28,29,30]. Nevertheless, some literature reported heightened self-cleaning and antibacterial potency against microorganisms for GO nanocomposite membranes [31,32,33,34].

Compared to GO-modified nanocomposite membranes, applications of graphene nanosheets incorporated polymeric membranes have not been considerably comprehended, where only a few pieces of literature are available. Some literature mentioned novel methodologies for enhancing their incorporation. In a recent study for direct contact membrane desalination applications, G. Grasso et al. (2020) have functionalized a commercial grade of Polyvinylidene fluoride (PVDF) with aromatic rings of styrene. Due to its adhesion potential, the styrene rings have acted as a base for the later fixation of graphene within the polymeric matrix. Results demonstrated that modified membranes manifested not only long-lasting salt retention but also greater stability [35]. In another work conducted by Mohamed Bayati et al. (2020) for ceramic membranes modification, laser-induced graphene was coated on microporous ceramic membranes’ substrate [36]. Results confirmed the firmly bonded graphene layer with a promising potential for many wastewater applications. For scale-up of graphene membranes, pristine graphene-coated hollow fiber membranes were also synthesized via the sacrificial layer-assisted CVD approach [37]. To cope with the issue of unintended defects of graphene membranes mass production, Siying Lia et al. (2018) suggested a novel route to prepare scalable CVD graphene polymer membranes through the formation of a polysulfone supporting layer [38]. The authors claimed that their nanocomposite membranes could pave the way for the commercialization of graphene membranes for desalination applications. Herein, we shed light on bulk modification of polyphenyl sulfone ultrafiltration membranes with graphene. It should be mentioned that the application of pristine graphene has not comprehensively been covered in literature as a filler in the casting solution due to the hydrophobic nature of the graphene, making its homogeneous blending with hydrophilic polymer impractical. Therefore, to conquer this challenge, Gum Arabic-Graphene (GGA) was prepared and used for the first time as an additive for the enhancement of the PPSU membrane properties. The presence of GGA could act as a pore-forming agent as well to enhance the surface and internal structure of membranes along with surface hydrophilicity characteristics. This very characteristic of Gum Arabic was attributed to the presence of the polysaccharide bond in its structure, which was the main reason for the improvement of the hydrophilic character of the membrane and its performance. Facilely, this would endow a better-dispersed nanostructure within the PPSU cast membranes. Ultimately, the resulting PES/GGA membrane performance could show a more reliable performance than pristine PES.

In this context, a low content ratio (0, 0.05, 0.1, 0.15, 0.2, and 0.25 wt.%) of Gum Arabic-Graphene was employed to probe the influence of the additives on the morphological changes of nanocomposites and their permeability characteristics. For performance evaluation, sodium alginate as an organic foulant model was harnessed during the study. A range of characterization tools has been employed for achieving this aim.

## 2. Experimental Section

### 2.1. Materials

Polyphenylsulfone, Ultrason^®^ P (PPSU) with an average molecular weight of 48,000 and transition temperature Tg = 220 °C was donated by BASF, Ludwigshafen, Germany. *N*-methyl-2- pyrrolidinone (NMP) was employed as an organic solvent. Graphite powder (332461) and Gum Arabic (G9752) were purchased from Sigma-Aldrich (Petaling Jaya, Malaysia). Sodium alginate NaAlg (12–80 kDa) from Brown algae as polysaccharide was obtained from Sigma-Aldrich.

### 2.2. Synthesis of Graphene Nanosheets

A simple green method was employed to produce a few layers of graphene through exfoliating graphite with Gum Arabic (GA)–Graphene (G) solution was first prepared by dissolving 10 mg mL^−1^ of graphite sheet into a volume of 800 mL of Gum Arabic in water. The best conditions are considered to the optimum production of graphene, and the exfoliation medium was fabricated prior to the mixing by blending 10 mg mL^−1^ of GA at 30 °C. The second stage consisted of a centrifuging process at 3000 rpm for GGA separation from the unexfoliated graphite. Then the obtained supernatant was washed by using filtration of several stages before freezing to drying [39].

From the statistical length distribution of Gum Arabic-Graphene observed by the TEM test, it was obtained that the mean lateral size of G after exfoliation was 450 nm, and typical Raman spectra of GGA and graphite were reported in previous work results [39]. FTIR spectra for GGA were also presented by Z. Ismail et al. [39].

### 2.3. Fabrication of Nanocomposite Ultrafiltration Membranes

All GGA/PPSU membranes were synthesized through the non-induced phase separation technique. A range from 0 to 0.25 wt.% GGA content was employed during the fabrication. The composition of the nanocomposite membranes is given in Table 1 below. The PPSU polymer was firstly died at 50 °C overnight to remove the moisture content. A 15 wt.% of the polymer was then dissolved in NMP solvent by stirring at 30 °C inside a sealed flask. The desired amount of GGA was added to the casting solution under sonication to achieve a homogenous solution. An appropriate amount of the degassed casting solution was poured on a glass substrate and cast via an automated casting knife with a 180-micron clearance gap. The resultant thin film was then directly immersed in a water bath for coagulation. Final nanocomposites were finally rinsed thoroughly by deionized water and stored for further characterization. 

### 2.4. Membrane Characterization

Membrane performance was evaluated through a crossflow testing rig having an active membrane area of 12.6 cm^2^ at 20 ± 0.5 °C. Initially, all membranes were compacted using deionized water at 0.5 MPa for 30 min. Following that, the pressure was lowered to 0.4 MPa, and the pure water flux was recorded. For the evaluation of the control and nanocomposite membranes performance, a concentration of 50 ppm NaAlg, as a feed solution, was passed through each membrane, and their flux decline was monitored while retention values were determined via Equation (1), below:R (%) = (1 − C_p_/C_f_) × 100(1)
where C_p_ and C_f_ are the NaAlg concentration in the permeate and feed solutions, respectively; the concentration of both feed and permeate was determined by the Total Organic Carbon analyzer (TOC-L, Shimadzu, Kyoto, Japan). Triplicate experiments with fresh membrane samples were used to present the measurement of the membrane performance, and the average value was taken into account for each sample.

Field Emission Scanning Electron Microscopy (TESCAN VEGA3 LM Oxford instruments, XMan, Prague, Czech Republic) was harnessed for observing the top surface and cross surface morphology of the nanocomposite membranes at an accelerating voltage of 30 kV.

Atomic force microscopy (AFM- SPM AA300 Angstrom Advanced Inc, Boston, MA, USA) imaging was employed to scan the surface topography of the composite’s membrane. Moreover, 2D and 3D images were obtained, and roughness parameters along with approximate mean pore size were determined using IMAGER 4.31 software with appropriate silicon tip.

For hydrophilicity measurements, an optical instrument (110-O4W CAM, Tainan, Taiwan) was employed to detect the water contact angle of the samples. In this method, a 3 μL of a deionized water droplet was placed onto the membrane surface using a microliter syringe. The profile of the water droplet on the surface was captured by an optical subsystem with a digital camera. At least three samples for each membrane were considered, and an average value of five locations for each sample was taken.

To quantify and verify functional groups of membranes along with possible molecular bonds between chemical compounds, a Fourier-Transform Infrared (FTIR) spectroscopy (Bruker Tensor 27 IR) was employed to rapidly identify the samples. The FTIR spectra were recorded between 400 cm^−1^ and 4000 cm^−1^ wavenumbers.

Thermal gravimetric analysis (TGA) was conducted (via TA Instrument-Q600SDT) to investigate the thermal decomposition of synthesized membranes; 5 mg of each membrane sample was placed inside the combustion chamber, and the samples were then heated in the presence of air up to 900 °C, using (heating rate 10 °C/min and airflow rate 10 mL/min).

## 3. Results and Discussion

### 3.1. Membrane Morphology 

Both Figure 1 and Figure 2 depicted the surface and cross-sectional morphologies of all synthesized membranes. As shown in Figure 1, all pure and GGA/PES membranes showcased a smooth, active layer with no particular GGA aggregates observed on the surface of the membranes. As revealed by the SEM images of the nascent PPSU membrane (MG1) in Figure 2, a well-formed finger-like microporous structure was formed within the entire membrane structure from the top to the bottom layer of the membrane. Noticeably, the pore walls were relatively thin, with a trivial sponge structure distributed randomly along with a thick skin layer at the top surface of the microporous membrane structure. This is confirmed with other intrinsic PPSU membranes morphology observed in preceding literature [40]. With the increasing loading content of impregnated GGA within the PPSU polymeric matrix, gradual changes in the morphology started to be seen. More obvious sponge-like structures started to appear with increasing the GGA content from 0.05 to 0.25 wt.%. As shown, nanocomposite membranes prepared using 0.05 wt.% (MG2) exhibited an almost similar structure to that of pristine PPSU membrane except for slightly wider pore walls with a more sponge structure. Increasing the GGA content has induced a denser structure membrane (MG3), whereas wide macropores began to be formed at the bottom of the membrane. These wide macropores tended to get bigger and aligned from the top to the bottom of the membrane (MG4). The pores’ lengths and their density have decreased. Higher GGA content (0.2 wt.%) have effectively changed the membrane morphology (MG5), where the finger-like structure considerably disappeared, and uneven big macrovoids were generated throughout the membranes. Undoubtedly, the role of the GGA ratio was more significant when using 0.25 wt.%. The lower half of the nanocomposite membrane (MG6) manifested a tiny finger-like morphology supported on a spongy support layer formed at the bottom half. Smaller independent macrovoids have been observed at this bottom half of the MG6 nanocomposite membranes. Observed morphological changes upon increasing the nano additives content are logically accepted since higher GGA content could induce greater dope solution viscosity. Higher casting solution viscosity is capable of inducing lower mixing-demixing between solvent and nonsolvent during the phase separation process [41]. Indeed, a denser active layer with a more sponge-like structure could be formed, as witnessed in this work.

### 3.2. Atomic Force Microscopy (AFM)

Surface topography and mean pore size of neat and nanocomposite membranes were scanned by AFM, and results were depicted in Figure 3 and Table 2, respectively. All microscopic observations were obtained under ambient conditions. In Figure 3, the shading on the 3D images corresponded to heights (peaks), while dark spots referred to low height (valleys). There was no clear correlation in the mean roughness value between the nanocomposite membranes prepared at different levels of GGA content. However, the mean surface roughness of pristine PPSU membrane manifested the lowest value (4.11 nm) besides maximum mean pore size (143.9 nm) amongst other GGA-modified nanocomposite membranes synthesized at disparate GGA contents. Impregnating 0.05 wt.% of GGA within the PPSU polymeric matrix has induced a significant variation in the nanocomposite membrane topography, where a greater number of small peaks/valleys have been generated. Herein, about a four-fold (16.5 nm) average roughness value was observed in comparison to the neat membrane. In the meantime, the mean pore size was smaller by about one-third (88.89 nm). Higher content (0.1 wt.%) of GGA showcased almost comparable mean roughness (15.3 nm) to that of MG2, whereas about 101.7 nm as an average mean pore size was recorded for this membrane (MG3). Raising the GGA content to 0.15, 0.2, and 0.25 wt.% had revealed uneven declined behavior in the average roughness, but still higher than the pristine membrane, and recorded 4.4, 5.2, and 12.9 nm for MG4, MG5, and MG6 nanocomposite membranes, respectively. In parallel, a decline in the mean pore size continued to be observed and was almost 95, 76, and 62 nm, as tabulated in Table 2. This enhancement in the texture of the modified membranes confirmed the excellent compatibility between GGA and organic PPSU polymer [40].

### 3.3. Hydrophilicity Measurements

Surface characteristics, including hydrophilicity, are one of the predominant features that are directly related to membrane water flux and surface fouling behavior. Contact angle measurements were employed to speculate all synthesized membranes’ hydrophilicity nature.

Figure 4 illustrates the contact angle values versus membrane composition. As shown, there was a noticeable decline in the contact angle value upon impregnating GGA nanosheets at 0.05 wt.%, from 69° for the neat PPSU membrane to about 52°. Following that ratio, a further amount of GGA incorporation has induced a further decline in the contact angle to about 51°, 50°, and 49° for nanocomposite membranes prepared with 0.1, 0.15, and 0.2 wt.%, respectively. These gradually diminished values could be stemmed from the desirable hydrophilic nature of the versatile hydrophilic functional groups (e.g., OH and COOH) at the surface and edge of the GGA. This confirms the potential of GGA to boost the wettability of the modified membranes by enhancing their hydrophilicity. However, nanocomposite fabricated at higher GGA content (0.25 wt.%) exhibited a slightly higher contact angle value (55°). This could be attributed to the weak dispersion and some GGA agglomeration in the PPSU polymeric matrices at a high loading ratio which diminished the surface/volume ratio of the nanosheets.

### 3.4. Fourier-Transform Infrared (FTIR) Spectroscopy 

Among the wide range of available characterization tools, FTIR has a special consideration as a powerful technique for the determination of versatile, functional groups on the membrane surface alongside their possible molecular bonds. FTIR spectra of synthesized pristine PPSU and PPSU-GGA membranes were given in Figure 5. Unsurprisingly, the infrared of the neat membrane exhibited the typical spectra of PPSU. The symmetric stretching characteristic absorption peak of the O=S= group in PPSU was noticed at about 1165 cm^−1^, while the characteristic peak of infrared anti-symmetric contraction of the O=S=O functional group was located at 1242 cm^−1^. Besides, there were clear characteristic peaks (1492 cm^−1^ and 1589 cm^−1^) attributed to the benzene ring stretching vibration (C=C). Comparing with GGA spectra reported in previous work, which appeared at 1638 cm^−1^ [39] and the control PPSU membrane as well as the composite membrane infrared spectra, it could be noticed that the absorption peak of the nanocomposite membrane showcased a larger width in the 3420 cm^−1^. This refers to the enhanced hydrophilicity of the PPSU membrane following the incorporation of GGA within the polymeric matrices [42].

### 3.5. X-ray Diffraction (XRD) Analysis of the Nanocomposite Membranes

The XRD patterns of pristine and graphene-modified PPSU membranes were illustrated below in Figure 6. The diffraction peak shown in all nanocomposite membranes at 26.6 corresponds to the (002) reflection of the graphene nanosheets [43,44]. Interestingly, although all GGAmodified PPSU membranes manifested the same peak, their peak intensity disclosed a proportional magnitude with each increment in the GGA content within the polymeric matrix. This confirms the successful incorporation of GGA at various loading weights employed in the current work. 

### 3.6. Influence of Graphene Content on the Thermal Stability of PPSU Membranes

TGA analysis of the pristine and GGA-modified membrane is illustrated in Figure 7. As shown, GGA-PPSU mixed matrix membrane manifested a trivial decline in the amount of mass before 150 °C. Initially, this weight loss was recorded below the onset decomposition temperature of PPSU and was ascribed to the moisture content in the membranes and/or decomposition of various functional groups. Pristine PPSU manifested a slightly greater decline (10%) than other nanocomposite membranes, which disclosed less decline (3–8%) at the same temperature. The significant weight loss due to PPSU polymer decomposition initiated at 150 °C and continued to about 250 °C, where at this range, all hydrocarbons decompose. Meanwhile, GGA-mixed matrix membranes exhibited greater thermal stability at the same condition. The onset decomposition temperature shifted from 150 °C to 170 °C when using only 0.05 wt.%, while this shift witnessed further development to about 180 °C when impregnating higher GGA content (0.25 wt.%). Moreover, the final decomposition temperature was 280 °C compared to 250 °C for the pristine PPSU membrane. These observations disclose that GGA could impart better thermal stability for the PPSU membranes by reducing the polymer proportion, diminishing the shift of the molecular polymer chain. Indeed, the interaction amongst GGA and PPSU polymeric chains made the fracture polymer chain require greater energy [40].

### 3.7. Evaluation of Membranes Performance with Sodium Alginate

The water permeation flux for the control PPSU membrane was compared with that of the GGA membrane. A representative organic solute (NaAlg) was used for the performance evaluation, as seen in Figure 8. A clear improvement was noticed in the pure water permeability characteristics of all UF nanocomposite membranes compared to that of the pure PPSU membrane. Under 0.4 MPa applied operating pressure, the PWF of the PPSU membrane was almost (40 ±1 L/m^2^·h). Compared to that of the PPSU membrane, the addition of 0.05 wt.% GGA has raised the PWF by one-third to about 60 ± 1 L/m^2^·h. The presence of a further amount of GGA in the polymeric matrix has induced a substantial flux enhancement (115 ± 2 L/m^2^·h) as witnessed by the membrane modified with 0.1 wt.% GGA. With a higher amount up to 0.15 wt.% of GGA, the membrane manifested the greatest PWF among other membranes, which recorded 119 ± 3 L/m^2^·h. More likely, this enhancement in the PWF stemmed from the imparted hydrophilic nature of GGA functional groups, which facilitated the diffusion of water molecules through the nanocomposite membrane. Beyond this GGA content, a slight decline in the flux started to be seen, where 114 ± 4 L/m^2^·h PWF was recorded for the nanocomposite modified with 0.2 wt.% GGA. This flux decline continued gradually to reach about 94 ± 4 L/m^2^·h when the highest amount of GGA (0.25 wt.%) was utilized in the synthesized membrane. It is worth mentioning here that even though the MG5 nanocomposite membrane had a lower contact angle than MG4, MG3, and MG2, their PWF was slightly lower. This indicates that the permeation characteristics of any membrane are a complex matter and not always associated with contact angle value. Other surface properties such as pore size, porosity, charge, and roughness can all interplay to determine the overall performance of the membrane. Win this context, NaAlg solute flux exhibited a similar trend, and the relative flux decline of the membranes was 40.5%, 11.5%, 38.4%, 31%, 28.1%, and 23% for the MG1, MG2, MG3, MG4, MG5, and MG6, respectively.

As illustrated in Figure 9, all nanocomposite membranes showcased a growing retention potential for NaAlg up to 0.15 wt.% GGA content, where beyond this content, a decline in the retention values was observed. After adding 0.05 wt.% GGA within the polymeric matrix of the membrane, it induced a higher rejection (74%) against sodium alginate compared to about 62% for pristine PPSU membrane. Further GGA content (0.1 and 0.15 wt.%) resulted in membrane matrices capable of achieving 82% and 88% retention for the sodium alginate, respectively. However, lower retentions were associated with membranes prepared with the highest GGA contents, which recorded about 72% and 48% for MG5 and MG6, respectively. Interestingly, the MG6 membrane revealed a lower rejection than pristine PPSU, even though it had a much smaller mean pore size. Probably this suggests that there were some defects formed in the MG6 matrix induced by the higher GGA content. Results disclosed that the mean pore sizes of MG5 and MG6 had the lowest values, about 76 and 62 nm, respectively. Moreover, their cross-section images manifest denser structures compared to other membranes. Therefore, it is logically accepted that these membranes should have a lower flux due to the high solution viscosity of the membranes. However, these flux values were still higher than the control membrane since they could have greater hydrophilicity and roughness where any membrane’s flux and retention behavior rely on a combination of surface characteristics. In the meantime, both membranes are supposed to exhibit higher retention values compared to all other membranes since they have the smallest mean pore size. However, in this work, they did not show higher retentions, which made us think some defects were created at higher GGA contents membranes and caused these declined retention values due to NaAlg passage through these defects.

Finally, a comparison between the membrane performance results of Gum Arabic-Graphene-modified polyphenylsulfone presented in the current work and selected from the literature was presented in Table 3. The most significant specifications of the nanocomposite membranes, such as contact angle, pore size, and porosity, were also illustrated in Table 3. The PPSU-GGA membranes have a logical solution permeation flux and solute removal efficiency (%) in comparison with those membranes selected from the literature. Moreover, it can also be seen from Table 3 that the Gum Arabic with loading amount of 1.5% in casting solution of 18% polysulfone reported by P.V. Chai et al. [45] has a contact angle of 56.74°, while in the current research, the contact angle was 50°. Whereas 3% of Gum Arabic embedded with 16% polysulfone by Manawi et al. [46] had a contact angle of 40.7° with permeation flux of 120.3 (L.m^−2^.h^−1^), which had better results than the present work due to the higher content amounts of Gum Arabic compared to the present work. According to what was mentioned here, it can be concluded that the amount of Gum Arabic should be optimized in order to present the highest values of membrane performance.

## 4. Conclusions

GGA-modified PPSU membranes for ultrafiltration applications were developed in the current study. Versatile characterization tools have been harnessed to assess the synthesized membranes quantitatively and qualitatively. Surface topography, surface and cross-sectional morphologies, thermal stability and crystalline structure, molecular structure, and wettability of all membranes were systematically identified. The cross-section morphologies revealed a huge transformation from finger-like to microporous and finally dense structures upon increasing the GGA content; also, slightly higher surface topography, smaller mean pore size, and higher thermal stability. The potential performance of the membranes was compared against NaAlg as an organic polysaccharide model. Unlike many preceding works of literature which employed high GGA content in the polymeric matrix, a moderate (up to 0.25 wt.% of GGA) additives content was found to be sufficient in this work. Results showcased that a membrane prepared using 0.15 wt.% GGA could bestow maximum permeation characteristics (119 ± 3 L/m^2^·h) and retention potential (88%), while optimum relative flux (11%) was obtained when using the minimum amount of GGA (0.05 wt.%). Besides, although modified membranes revealed a comparable hydrophilic nature (contact angle within the fifties), their performance was slightly dissimilar. This indicates that a permeation characteristic of any membrane is a complex matter and not always associated with contact angle value. Other surface properties such as pore size, porosity, charge, and roughness can all interplay to determine the overall performance of the membrane.

## Figures and Tables

**Figure 1 membranes-11-00542-f001:**
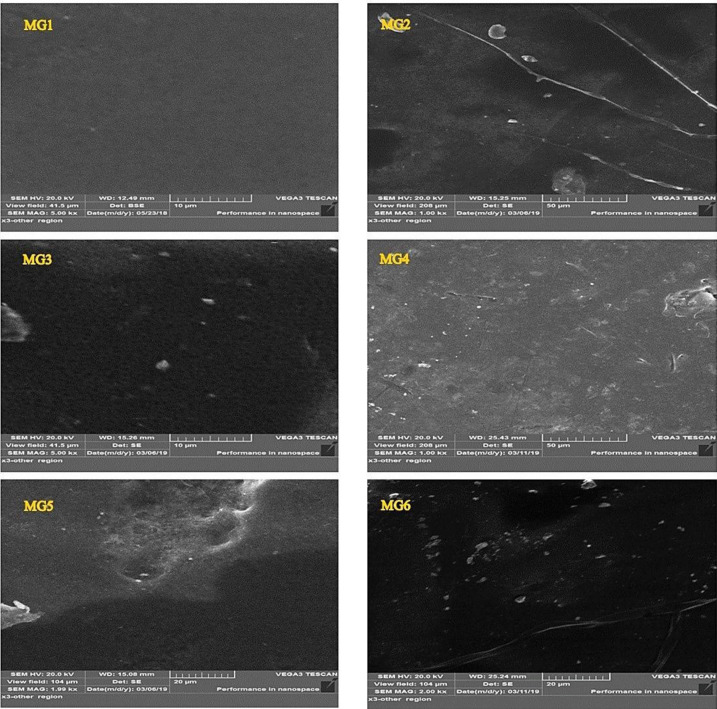
The FESEM images of the cross-sections of pristine PPSU and the nanocomposites at different weight percentages 0.05, 0.1, 0.15, 0.2, and 0.25 wt.% GGA.

**Figure 2 membranes-11-00542-f002:**
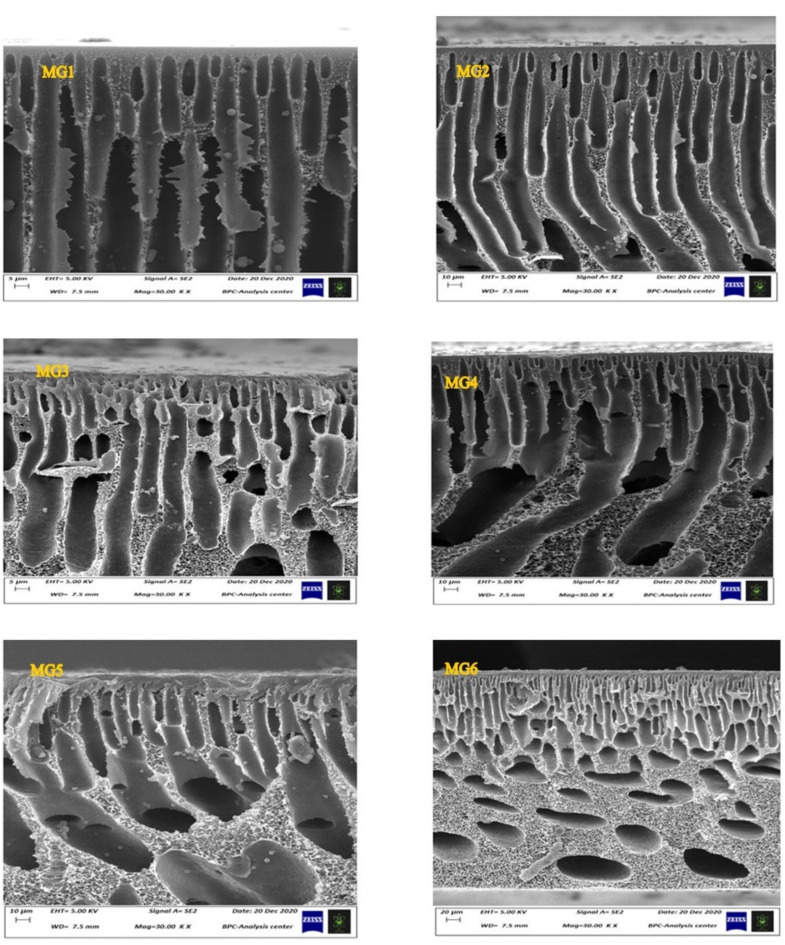
The FESEM images of the cross-sections of pristine PPSUs and the nanocomposites at different weight percentages 0.05, 0.1, 0.15, 0.2, and 0.25 wt.% GGA.

**Figure 3 membranes-11-00542-f003:**
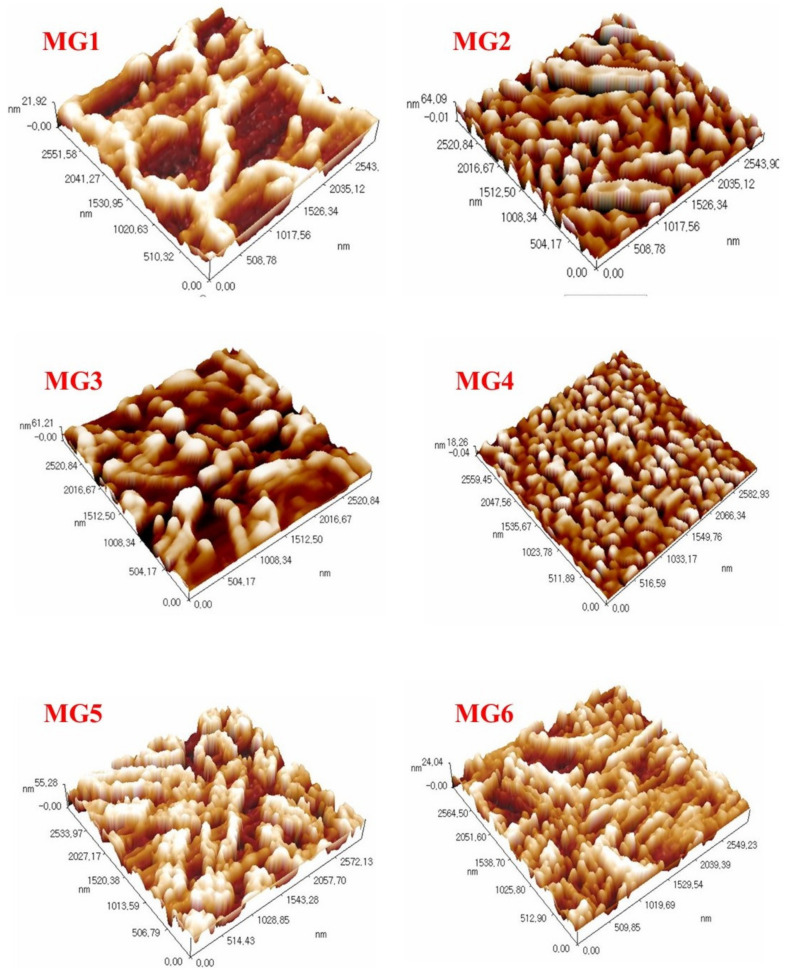
AFM images for the top membrane surface.

**Figure 4 membranes-11-00542-f004:**
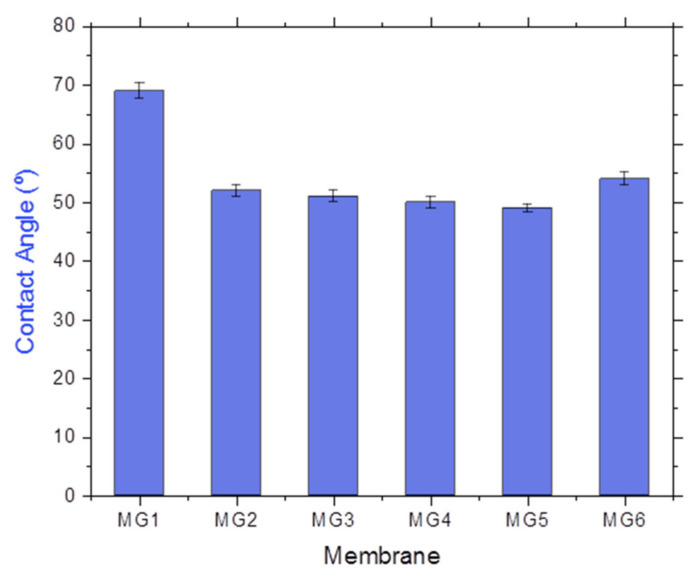
Contact angle measurements of synthesized membranes.

**Figure 5 membranes-11-00542-f005:**
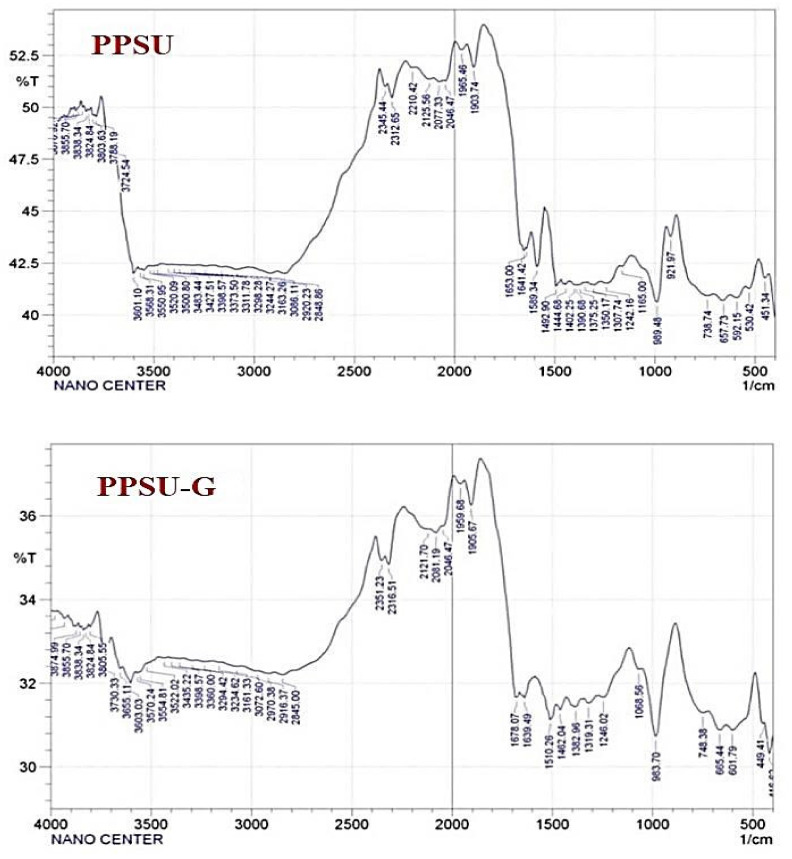
FTIR spectra of synthesized pristine PPSU and PPSU-GGA membranes.

**Figure 6 membranes-11-00542-f006:**
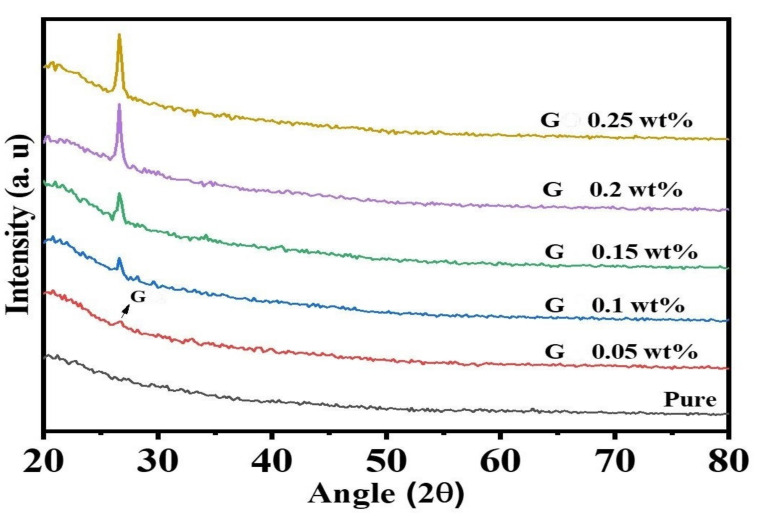
*X*-ray Diffraction (XRD) analysis of the membranes.

**Figure 7 membranes-11-00542-f007:**
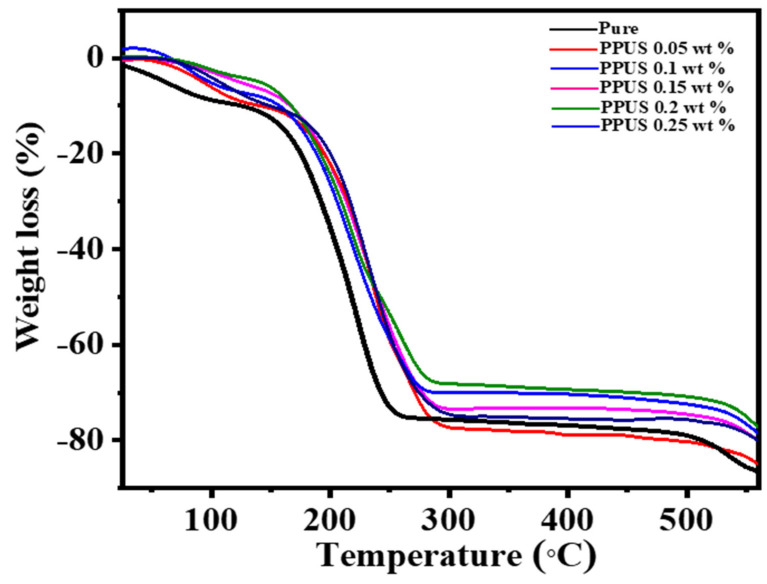
Thermogravimetric analysis of PPSUs membranes at different loading levels of GGA.

**Figure 8 membranes-11-00542-f008:**
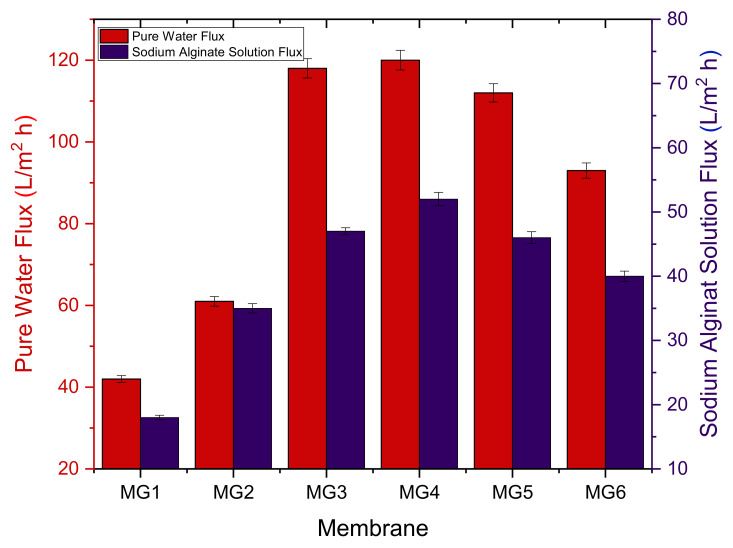
Pure water flux and NaAlg solute alginate solution flux of synthesized membranes.

**Figure 9 membranes-11-00542-f009:**
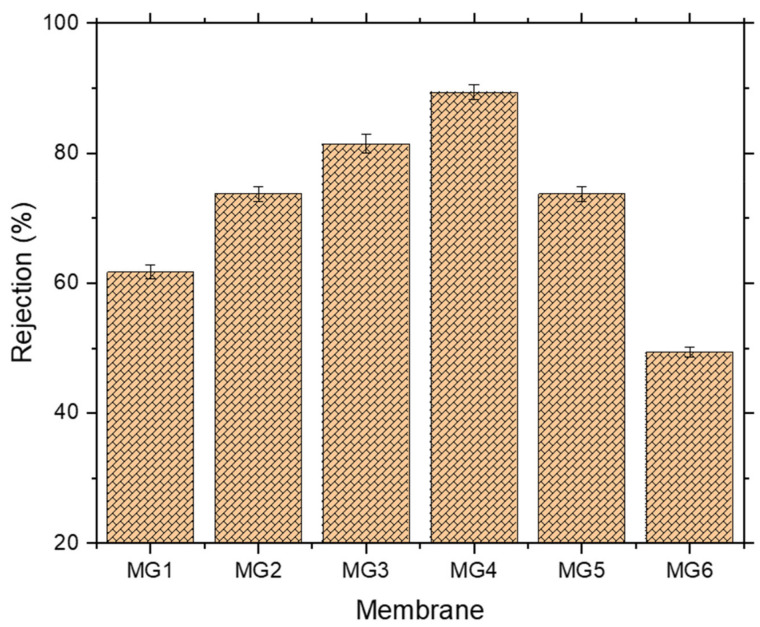
Separation performance of synthesized membranes.

**Table 1 membranes-11-00542-t001:** Composition of the nanocomposite membranes.

Membrane Code	PPSU wt. %	G wt. %
MG1	15	0
MG2	15	0.05
MG3	15	0.1
MG4	15	0.15
MG5	15	0.2
MG6	15	0.25

**Table 2 membranes-11-00542-t002:** Membrane average roughness and mean pore size.

Membrane Code	Average Roughness (nm)	Mean Pore Size (nm)
MG1	4.11	143.91
MG2	16.5	88.89
MG3	15.3	101.71
MG4	4.42	95.57
MG5	5.27	76.78
MG6	12.9	62.79

**Table 3 membranes-11-00542-t003:** Demonstrates the comparison of this work with the recently published work selected from the literature.

Mixed Matrix Membrane		Porosity (%)	Mean Pore Size (nm)	Contact Angle (°)	Rejection (%)	Permeation (L.m^−2^.h^−1^)	Ref.
Casting Solution (wt.%)	Nanoparticles Concentration						
15% Polyphenylsulfone and 85% *N*-Methyl-2-pyrrolidinone	0.15% Gum Arabic-Graphene	NA	95.57	50	88% Sodium Alginate	82.11	This Work
18% Polysulfone and 82% *N*-Methyl-2- pyrrolidone	1.5% Gum Arabic–0.6% Graphene Oxide	78.37	20.76	56.74	96.34% Humic Acid	63.55	[45]
16% Polysulfone and 84% Dimethylacetamide	3% Arabic Gum	70.3	37	40.7	% 80 BSA *	120.3	[46]
Cellulose Acetate, Vinyl Triethoxysilane, Graphene and Dimethyl Formamide	8% Gum Arabic	NA	NA	56	97.6% Pb(II) ion	8.6	[47]
20% Polyamide 6,6 and 80% Formic Acid	0.8% Silver-Graphene Oxide	67.96	8.26	35.28	89.8% BSA and 88.9% Congo Red	NA	[48]
21% Polyethersulfone, 1% Polyvinylpyrrolidone and 78% Dimethyl Sulfoxide	0.5% Graphene Oxide	80.6	14.59	39.21	99.7% Acid Black 210 and 99% Rose Bengal	116.5	[29]
17.5% Polyphenylsulfone, 1% Polyvinylpyrrolidone and 81.5% *N*-Methyl-pyrrolidone	0.5% Graphene Oxide	80	10.6	45	94% BSA and 88% Pepsin	171	[40]
20% Polyethersulfone, 1% Polyvinylpyrrolidone and 79% Dimethylacetamide	0.5% Graphene Oxide	83.1	4.5	53.2	96% Direct Red 16	NA	[28]
15% Polysulfone and 85% *N,N*-Dimethylformamide	2% Graphene Oxide	82.1	8.7	54.8	83.65% Arsenate	41.18	[49]
18% Polyvinylidene Fluoride and 82% *N*-*N*-Dimethylacetamide	0.1 g.L^−1^ Oxidized Multi-Walled Carbon Nanotubes	45.33	8.09	71.77	81.94% Turbidity, 86.3% Color, and 100% TSS of Palm Oil Mill Effluent	131.97	[50]

* BSA: Bovine Serum Albumin.

## Data Availability

Not applicable.

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
