# Peer review of "Fabrication of Gum Arabic-Graphene (GGA) Modified Polyphenylsulfone (PPSU) Mixed Matrix Membranes: A Systematic Evaluation Study for Ultrafiltration (UF) Applications"

_membranes, 2021, doi:10.3390/membranes11070542_

Round 1

Reviewer 1 Report

The authors of the manuscript entitled “Fabrication of gum Arabic–graphene (GGA) modified poly-phenylsulfone (PPSU) mixed matrix membranes: a systematic evaluation study for ultrafiltration (UF) applications” present an interesting work, with a well-thought-out experimental process and membrane characterization techniques. Some key points need to be clarified prior to publication:

Extensive English editing is required.

The lower rejection of MG5 and MG6 membranes compared to MG4 is attributed to membrane defects? Wouldn’t this lead to higher water fluxes? Wouldn’t something show-up in the FESEM images?

The origin of the error bars should be stated for each measurement, is it repeat of the measurement or repeat of the experiment? Was the error propagation formula used for the error bars of indirectly measured properties (like rejection)?

Even though standard deviation of most of the measurements is presented, statistical analysis of the results (like a simple t-test) will clearly state the statistical significance of the effect of GA-graphene nanoparticles addition on membrane properties.

Table 3 should be further discussed to illustrate the advantages of using the proposed membrane fabrication technique compared to literature.

Author Response

Cover Letter Dear Editor I would like to submit the revised form of the manuscript entitled “"Fabrication of gum Arabic–graphene (GGA) modified polyphenylsulfone (PPSU) mixed matrix membranes: a systematic evaluation study for ultrafiltration (UF) applications" for consideration for publication in the Membranes. This article is revised according to the reviewer comments. All answer to the reviewers comment was highlighted in the red colour in the manuscript. I appreciate the effort of the reviewers to improve our article, thank you. Please answer to the reviewers' comments as in the following report. Your consideration for this manuscript with revised form is highly appreciated. Sincerely Prof. Dr. Qusay F. Alsalhy Membrane Technology Research Unit Chemical Engineering Department University of Technology, Alsinaa Street No. 52 Baghdad, Iraq Email: [email protected] [email protected] Answer to the reviewers Comments Reviewer 1 The authors of the manuscript entitled “Fabrication of gum Arabic–graphene (GGA) modified poly phenylsulfone (PPSU) mixed matrix membranes: a systematic evaluation study for ultrafiltration (UF) applications” present an interesting work, with a well-thought-out experimental process and membrane characterization techniques. Some key points need to be clarified prior to publication: 1) Extensive English editing is required. Ans: The manuscript English has been revised as suggested by the reviewer 2) The lower rejection of MG5 and MG6 membranes compared to MG4 is attributed to membrane defects? Wouldn’t this lead to higher water fluxes? Wouldn’t something show-up in the FESEM images? Ans. Thanks for raising this valuable comment. Results disclosed that the mean pore sizes of MG5 and MG6 had the lowest values, about 76 and 62 nm, respectively. Also, their cross-section images manifest denser structures comparing to other membranes. Therefore, it is logically accepted that these membranes should have lower flux due to the high solution viscosity of the membranes. However, these flux values were still higher than the control membrane since they could have greater hydrophilicity and roughness where any membrane's flux and retention behaviour rely on a combination of surface characteristics. In the meantime, both membranes supposed to exhibit higher retention values comparing to all other membranes since they have the lowest mean pore size. But in this work, they did not show higher retentions, which made us think some defects were created at higher GGA contents membranes and caused these declined retention values due to NaAlg passage through these defects. For the last point, when using FESEM, we only scan a small area and most probably would not be able to observe these defects. Therefore we added the following paragraph highlighted in red color: Results disclosed that the mean pore sizes of MG5 and MG6 had the lowest values, about 76 and 62 nm, respectively. Also, their cross-section images manifest denser structures comparing to other membranes. Therefore, it is logically accepted that these membranes should have lower flux due to the high solution viscosity of the membranes. However, these flux values were still higher than the control membrane since they could have greater hydrophilicity and roughness where any membrane's flux and retention behaviour rely on a combination of surface characteristics. In the meantime, both membranes supposed to exhibit higher retention values comparing to all other membranes since they have the lowest mean pore size. But in this work, they did not show higher retentions, which made us think some defects were created at higher GGA contents membranes and caused these declined retention values due to NaAlg passage through these defects. 3) The origin of the error bars should be stated for each measurement, is it repeat of the measurement or repeat of the experiment? Was the error propagation formula used for the error bars of indirectly measured properties (like rejection)? Ans: Error bars have been added to all graphs (contact angle, rejection and fluxes). Triplicate experiments with fresh membrane samples were used to present the measurement in this work. We added the following sentences to the characterization section: • For contact angle we added the following sentence: At least three samples for each membrane was considered and an average value of five locations for each sample was taken. • For rejection and fluxes: Triplicate experiments with fresh membrane samples were used to present the measurement of the membrane performance and the average value was taken into account for each sample. 4) Even though standard deviation of most of the measurements is presented, statistical analysis of the results (like a simple t-test) will clearly state the statistical significance of the effect of GA-graphene nanoparticles addition on membrane properties. Thank you for your smart comment, statistical analysis of the results by t-test as an example was not taken in account in the current work. Of course it will be considered in the next research works of our research group. 5) Table 3 should be further discussed to illustrate the advantages of using the proposed membrane fabrication technique compared to literature. Answer: Finally, a comparison between the membrane performance results of gum arabic - graphene-modified polyphenylsulfone presented in the current work and that selected from the literature was presented in Table 3. The most significant specification of the nanocomposite membranes such as contact angle, pore size and porosity were also illustrated in Table 3. The PPSU-GGA membranes have a logical solution permeation flux and solute removal efficiency (%) in comparison with those membranes selected from the literature. Moreover, it can also be seen from Table 3 that the Gum Arabic with loading amount of 1.5% in casting solution of 18 % polysulfone reported by P.V. Chai et al., [45] has a contact angle of 56.74 °, while in the current research the contact angle was 50 °. Whereas, 3% of Gum Arabic embedded with 16% polysulfone by Manawi, et al., [46] has a contact angle of 40.7 ° with permeation flux of 120.3 (L.m-2.h-1) which was better results than the present work due to the high contant amount of Gum Arabic than the present work. According to what mention here it can be conclude that the amount of Gum Arabic should be optimized in order to present the highst values of membrane performance. Reviewer 2 The manuscript focused on the evaluation of GGA/PPSU mixed matrix membranes for ultrafiltration. The modified graphene improves membrane hydrophilicity and separation performance. The following are my concerns: 1 In introduction, please emphasize the advantage of using Gum-Arabic as modifier for graphene. Ans. : (line 108-119) The authors agree with the reviewer comment, please see the introduction section as we have considered your suggestion: " The presence of GGA could act as a pore-forming agent as well to enhance the surface and internal structure of membranes along with surface hydrophilicity characteristics. This own character of Gum Arabic was attributed to the presence of the polysaccharide bond in its structure which was the main reason for improvement of the hydrophilic character of the membrane and its performance. Facilely, this would endow a better-dispersed nanostructure within the PPSU cast membranes. Ultimately, the resulting PES/GGA membrane performance could show a more reliable performance than pristine PES." 2 Please include the property (size of graphene, chemical and morphology) of Graphene and modified graphene in the manuscript. Answer: We agree with the reviewer comment, therefore we added the following information in section 2.1. Synthesis of graphene nanosheets: From the the statistical length distribution of Gum Arabic-Graphene observed by TEM test it was obtained that the mean lateral size of G after exfoliation was 450 nm and typical Raman spectra of GGA and graphite were reported in the results of previous work [39]. FTIR spectra for GGA was also presented by Z. Ismail et. al., [39]. 3 What is the viscosity of different polymer solution? Please include in Table 1. Ans: Thanks for the reviewer, In this work authors are focusing on membrane modification with novel synthesized materials. Therefore, we cannot cover all tests in one work such as the dope solution viscosity characteristics. 4 Figure 1, the images are not clear. Please adjust the brightness or change the image. Ans: The brightness of the image has been adjusted 5 In Figure 5, please compare the spectra with the modified graphene. Answer: We agree with the reviewer comment, therefore please see the revised paragraph: "Comparing with GGA spectra reported in previous work which were a peaks appear at 1638 and 3460 cm-1 [39] and the control PPSU membrane as well as the composite membrane infrared spectra, it could be noticed that the absorption peak of the nanocomposite membrane showcased a larger width in the 3420 cm−1. This refers to the enhanced hydrophilicity of the PPSU membrane following the incorporation of GGA within the polymeric matrices [42]." 6. In Figure 6, please also add the spectra of graphene and modified graphene in this figure. 7. In Figure 7, please also add the TGA data for the graphene and modified grapheme Ans: Regarding comments nos. 6 and 7, we would like to the reviewer that the authors did not measure the X-ray Diffraction (XRD) analysis and Thermogravimetric analysis of graphene and modified graphene for avoid of repeating the results, as they were measured in the previous research [39] 6 Can the author show the change in MWCO before and after modification of the membranes? Ans: The authors have presented the mean pore size values for all membranes as could be seen in Table 2. Measuring the mean pore size is consider as an adequate alternative to MWCO. Reviewer 3 This problem is relevant for journal scope. The manuscript follows the formal regulations of MDPI journals. I suggest the acceptance after minor revision. Remarks, suggestions, question 1 The main contribution and novelty of the work are not identified. The scientific justification for this work should be added. Ans. : The contribution and novelty of the work have been clearly identified as could be seen in the introduction section (lines 108-119 ) "Authors claimed that their nanocomposite membranes could pave the way for the commercialization of graphene membranes for desalination applications. Herein, we shed the light on bulk modification of polyphenyl sulfone ultrafiltration membranes with Graphene. It should be mentioned that the application of pristine graphene has not comprehensively covered in literature as a filler in the casting solution due to the hydrophobic nature of the graphene makes its homogeneous blending with hydrophilic polymer impractical. Therefore, to conquer this challenge, gum Arabic–graphene (GGA) was prepared and used for the first time as an additive for enhancement of the PPSU membrane properties. The presence of GGA could act as a pore-forming agent as well to enhance the surface and internal structure of membranes along with surface hydrophilicity characteristics. This own character of Gum Arabic was attributed to the presence of the polysaccharide bond in its structure which was the main reason for improvement of the hydrophilic character of the membrane and its performance. Facilely, this would endow a better-dispersed nanostructure within the PPSU cast membranes. Ultimately, the resulting PES/GGA membrane performance could show a more reliable performance than pristine PES." 2 Please cite more papers from MDPI journals at the last two years in the similar topic of this research. Answer: more papers have been added and we already take the requested by the reviewer into account. Please see the highlighted references in red color in the list of references 3 Please add the Nomenclature part to the manuscript. Ans: Thanks to the reviewer, we have added the Nomenclatures at the end of the manuscript (after the conclusion section) 4 Please add some information about cut-off value. Ans: The authors have presented the mean pore size values for all membranes as could be seen in Table 2. Measuring the mean pore size is consider as an adequate alternative to MWCO. 5 Concentration polarization is a well-known phenomenon in the case of pressure-driven membranes. What do you think about concentration polarization in the case of your model? Ans: We agree with these comments. However, in this work Concentration polarization phenomenon does not fall within the objectives of the research and it will be taken into consideration during subsequent studies. 6 Please add information about mass and component balance. Ans: In this work mass and component balance does not fall within the objectives of the research and it will be taken into consideration during subsequent studies.

Reviewer 2 Report

The manuscript focused on the evaluation of GGA/PPSU mixed matrix membranes for ultrafiltration. The modified graphene improves membrane hydrophilicity and separation performance. The following are my concerns:

  1. In introduction, please emphasize the advantage of using Gum-Arabic as modifier for graphene.
  2. Please include the property (size of graphene, chemical and morphology) of Graphene and modified graphene in the manuscript.
  3. What is the viscosity of different polymer solution? Please include in Table 1.
  4. Figure 1, the images are not clear. Please adjust the brightness or change the image.
  5. In Figure 5, please compare the spectra with the modified graphene.
  6. In Figure 6, please also add the spectra of graphene and modified graphene in this figure.
  7. In Figure 7, please also add the TGA data for the graphene and modified graphene.
  8. Can the author show the change in MWCO before and after modification of the membranes?

Author Response

(The authors gave the same response as above.)

Reviewer 3 Report

This problem is relevant for journal scope. The manuscript follows the formal regulations of MDPI journals.

I suggest the acceptance after minor revision.

Remarks, suggestions, question

  1. The main contribution and novelty of the work are not identified. The scientific justification for this work should be added.
  2. Please cite more papers from MDPI journals at the last two years in the similar topic of this research.
  3. Please add Nomenclature part to the manuscript.
  4. Please add some information about cut-off value.
  5. Concentration polarization is a well-known phenomenon in the case of pressure-driven membranes. What do you think about concentration polarization in the case of your model?
  6. Please add information about mass- and component balance.

Author Response

(The authors gave the same response as above.)

Round 2

Reviewer 1 Report

All comments have been addressed by the authors

Reviewer 2 Report

The authors addressed all my concerns.